# The Rise of AI-Assisted Diagnosis: Will Pathologists Be Partners or Bystanders?

**DOI:** 10.3390/diagnostics15182308

**Published:** 2025-09-11

**Authors:** Riyad El-Khoury, Ghazi Zaatari

**Affiliations:** Department of Pathology and Laboratory Medicine, American University of Beirut Medical Center, Beirut 1107 2020, Lebanon; re70@aub.edu.lb

**Keywords:** digital pathology, artificial intelligence, whole-slide imaging, foundation models, diagnostic automation, pathologist-free diagnosis, clinical decision support

## Abstract

Over 150 years, pathology has transformed remarkably, from the humble beginnings of microscopic tissue examination to today’s revolutionary advancements in digital pathology and artificial intelligence (AI) applications. This review briefly retraces the evolution of microscopes and highlights breakthroughs in complementary tools and techniques that laid the foundation for modern surgical pathology, recently expanded into a new dimension with digital pathology. Digital pathology marked a pivotal turning point by addressing the longstanding limitations of conventional microscopy, paving the way for AI integration. AI now revolutionizes pathology workflows, offering unprecedented opportunities for automated diagnostics, enhanced precision, accelerated research, and advanced medical education. Despite widespread consensus on AI as complementary to pathologists, rare studies critically explore the feasibility of a fully autonomous, pathologist-independent diagnostic workflow. Given the rapid advancement of AI, it is timely to examine whether mature AI systems might realistically achieve diagnostic autonomy. Thus, this review uniquely addresses this gap by evaluating the feasibility, limitations, and implications of a disruptive, pathologist-free diagnostic model. This exploration raises critical questions about the evolving role of pathologists in an era increasingly defined by automation. Can pathologists adapt to emerging trends, maintain their central role in patient care, and leverage AI effectively, or will their traditional roles inevitably diminish? Could the continued advancement of AI eventually prompt a return of pathologists to their initial mid-19th century role as scientist scholars, removed from frontline diagnostics? Ultimately, we assess whether AI can independently sustain diagnostic accuracy and decision making without pathologist oversight.

## 1. A Brief Journey from Early Microscopes to Modern Anatomic Pathology: Foundations of a Digital Revolution

The origin of microscopy, much like pathology, was a gradual and multifaceted journey. One of the earliest advancements in optics was the invention of spectacles—lenses in a frame, used to improve vision—first referenced in 1299 in a manuscript from Florence and attributed to Salvino d’Armato degli (1285) [1]. These early lenses laid the groundwork for the later development of more sophisticated optical tools. While simple magnifying devices were used for centuries, the development of the first compound microscope (an instrument consisting of a suitable combination of lenses and mirrors) emerged between 1590 and 1610, marking the beginning of microscopic visualization of biological structures [2,3] (Figure 1).

Galileo Galilei is widely credited for inventing the first compound microscope in 1610 by experimenting with telescope lenses, although Dutch spectacle makers, Hans and Zacharias Janssen, had reportedly used telescope lenses to enlarge small objects as early as 1590 [1,4]. Galileo’s compound microscope, called “*occhialino*”, represented a reduced form of the Janssens’ telescope and inspired further optical advancements. One notable innovation was Cornelius Drebbel’s powerful microscope in 1621, which surpassed the capabilities of Galileo’s *occhialino* and marked a significant step forward in the evolution of microscopy [5]. Robert Hooke’s microscope (1665), John Marshall’s design (1700), and the Oberhauser–Lerebours microscope (1841), among other innovations in microscopy, contributed to the development of modern compound microscopes, enabling the study of cells and tissues at the microscopic scale (Figure 1) [2,6,7,8]. While microscopes transformed visualization, the advancement of pathology required additional breakthroughs in complementary tools and techniques. The introduction of the microtome in the 1830s enabled precise and reproducible tissue sectioning, and freezing microtomes in the 1870s facilitated the development of the frozen sections technique, which is still critical today for rapid intraoperative diagnosis [9,10]. Paraffin embedding, introduced by Edward Klebs in 1869, and formalin fixation by Ferdinand Blum in 1893 further improved and standardized tissue processing. Meanwhile, histological staining methods such as Franz Böhm’s hematoxylin staining in 1865 marked a pivotal moment, creating a method that remains a cornerstone of today’s histopathology [11,12] (Figure 1). Undoubtedly, hematoxylin and eosin (H&E) staining remains a diagnostic gold standard for its clarity, cost-efficiency, and wide applicability. A major leap occurred in 1941, when Albert Coons et al. developed immunofluorescence using fluorescein isothiocyanate-labeled antibodies [13,14]. Despite its utility in research, it was less suited for routine surgical pathology due to its reliance on fresh tissue, the need for fluorescent microscopes, and fluorescence fading. In the 1960s, enzyme-labeled antibodies enabled durable, visible staining suitable for routine diagnostics, laying the foundation for immunohistochemistry (IHC) [15]. IHC has since revolutionized diagnostics by allowing targeted antigen localization in tissues, complementing H&E in tackling complex diagnostic challenges [14,16,17]. The development of photomicrography in the mid-19th century, allowing microscopic images to be captured and studied, further solidified the role of microscopy in pathology [18,19].

By the mid-to-late 19th century, pathology began to emerge as a distinct medical discipline, though its practice was still closely tied to the clinical and surgical practice [20,21,22]. The influence of Karl von Rokitansky and Rudolf Virchow in the mid-19th century marked a turning point in pathology. Rokitansky, a proponent of gross anatomical pathology, centralized autopsy pathology, separating it from direct clinical practice [23,24]. Virchow integrated microscopy into autopsy studies, transforming pathology into a scientific field and a cornerstone of modern medicine [25,26,27]. This historical transition, from generalist scholar-clinicians to specialized diagnosticians, parallels the technological shifts that would transform pathology into a structured scientific discipline and pave the way for digital diagnostics [25,26,27].

For over a century, light microscopy has been the “gold standard” for pathological diagnosis and medical education. However, as diagnostic demands grew and technology advanced, the limitations of classical microscopy, such as reliance on physical slides, the difficulty of remote consultations or archiving, and the limited scalability, became increasingly evident. A pivotal advancement occurred in 1994 when James Bacus developed the BLISS (Bacus Laboratories Inc. Slide Scanner) system, which is the first commercial slide scanner. This innovation paved the way for the emergence of digital pathology, which addressed many of the constraints of traditional microscopy by converting traditional glass slides into high-resolution whole-slide images (WSIs) [28]. Over the past 30 years, slide scanning technology has undergone significant and continuous refinement, enhancing its image quality, scanning speed, and accessibility. Several recent reviews have compared the performance, diagnostic capabilities, and technical features of leading commercial scanners and associated software, offering valuable insights for readers interested in the practical implementation of digital pathology [29,30,31,32].

With digital pathology paving the way, artificial intelligence has emerged as a transformative force, promising to revolutionize how tissue samples are analyzed and interpreted. But what does this mean for the practice of pathology today? Does AI challenge or complement the expertise of pathologists? To explore these questions, we must first explore the current landscape of digital pathology and the transformative role of AI in reshaping the field.

## 2. Current Landscape of Digital Pathology and Key Advancements in AI

Microscopic examination has long enabled pathologists to assess cellular-level alterations, offering critical insights into diagnosis and prognosis. However, reliance on manual glass slides has inherent limitations. Traditional microscopy is time-consuming and susceptible to human error and variability among different investigators, potentially impacting diagnostic consistency. Moreover, physical slides present significant logistical challenges related to storage, transportation, and susceptibility to damage. Conventional microscopy also hindered widespread collaboration, as physical slides necessitated localized expert consultations and restricted accessibility for remote or global case sharing and teamwork.

The advent of digital pathology, which refers to the transformation of traditional glass slides using specific technologies and methods into high-resolution WSIs with associated metadata, has provided a revolutionary solution to those constraints [33,34]. Digital pathology enables real-time remote consultations, second opinions, and multidisciplinary discussions without the need to transport physical slides. It allows electronic and long-term storage on highly secure servers or cloud-based platforms, reducing the risk of physical damage or loss of slides, and offers unprecedented possibilities for workflow optimization through rapid retrieval of cases and integration with laboratory information systems (LISs). Importantly, digital pathology serves as the essential technological foundation for the application of AI in diagnostic workflows. By converting physical slides into high-quality digital data, it provides the standardized, scalable input required for training and deploying AI algorithms. In this sense, digital pathology is not only a modernization of slide handling but a prerequisite infrastructure for the development of AI-powered tools in pathology. Additionally, digital pathology supports automated measurement and annotation, improving diagnostic precision and consistency. It also enables the creation of virtual slide libraries for education and training, offering interactive and accessible learning experiences previously unattainable with conventional microscopy.

### 2.1. Technologies Driving Digital Pathology

Several technologies developed concomitantly with slide scanners have significantly enhanced the adoption and efficiency of digital pathology workflows. Key advancements include high performance computing with AI accelerator chips, robust image management systems often integrated with LIS to streamline workflows, improved deep learning architectures and reinforcement learning techniques, cloud computing and storage solutions, data compression algorithms allowing efficient transmission, storage, and processing of images without compromising image quality, secure communication protocols ensuring compliance with data privacy regulations for secure sharing of digital slides in telepathology and remote collaborations [35,36]. These advancements have not only optimized digital pathology workflows but have also paved the way for the integration of AI-assisted diagnostics [37,38]. Recent regulatory approvals and real-world studies underscore both the promise and the practical readiness of AI in pathology. Paige Prostate Detect (Paige AI Inc., New York, NY, USA; https://www.paige.ai/diagnostic-ai; accessed on 5 August 2025), the first FDA-cleared AI pathology tool designed to assist pathologists in detecting prostate cancer, demonstrated a statistically significant improvement in sensitivity, with a 7.3% reduction in false negatives, while being compatible across diverse scanner platforms [39,40]. Furthermore, the U.S. FDA has granted Breakthrough Device Designation to Paige’s PanCancer Detect (https://www.paige.ai/diagnostic-ai; accessed on 5 August 2025), an AI system intended to support cancer detection across multiple anatomical sites [41]. Large-scale NHS deployments under the ARTICULATE PRO initiative further illustrate the feasibility of implementing AI-powered pathology tools across varied clinical settings and patient populations [42]. Finally, MSIntuit™ CRC (Owkin Inc., New York, NY, USA; https://www.owkin.com/diagnostics/msintuit-crc; accessed on 5 August 2025) represents a new class of AI-based decision-support tools designed to triage colorectal cancer slides for possible microsatellite instability, helping to prioritize cases for downstream confirmatory analyses such as immunohistochemistry or next-generation sequencing, and thus optimizing diagnostic efficiency [43].

### 2.2. Artificial Intelligence: A New Frontier in Pathology

The advent of AI ushered in a new, transformative era in pathology. Development and implementation of increasingly powerful AI-based tools, relying primarily on deep learning models, offer solutions to some of the field’s most critical challenges, such as minimizing intra- and inter-observer interpretation variability, managing rising diagnostic workloads, mitigating diagnostic delays, and integrating complex data from diverse sources into a unified diagnostic framework.

Task-specific AI models

Visual inspection of histopathology slides can be hindered by the subjective nature of assessing histological features such as mitotic figures quantification, atypia evaluation, tumor-infiltrating lymphocytes scoring, and Ki67 proliferation index estimation. These and similar tasks often suffer from variability among pathologists. Moreover, many microscopic patterns can be imperceptible to the human eye.

Researchers are leveraging task-specific deep learning-based models with varying degrees of success to address a diverse range of tasks, including cancer subtyping and grading [44,45,46], recurrence prediction [47,48], metastasis detection [49,50], survival and response-to-treatment prediction, tumor site of origin prediction, mutation prediction and biomarker screening [51,52,53], and more. Performances across these applications have, in several cases, been demonstrated as similar to, or potentially better than, those of experienced pathologists [54,55].

For example, in the CAMELYON16 (Cancer Metastases in Lymph Nodes Challenge 2016) challenge, a set of algorithms for detecting breast cancer metastasis into lymph nodes was developed by various research groups around the world. The best algorithm achieved an area under the curve (AUC) of 0.994, outperforming a panel of 11 expert pathologists (mean AUC, 0.810; range, 0.738–0.884; *p*  <  0.001) in a time-constrained manner [49]. A deep learning system for Gleason scoring in prostate cancer grading has similarly shown a significantly higher diagnostic accuracy of 0.70 (*p* = 0.002; 95% CI: 0.65–0.75) when compared to experienced pathologists with a mean accuracy of 0.61 (95% CI: 0.56–0.66) [44]. Additionally, a fully trained Inception v3 deep learning model achieved an AUC of 0.95 for distinguishing lung adenocarcinomas (LUADs) from lung squamous cell carcinoma (LUSC), outperforming specialized pathologists [52]. In the same study, LUAD mutation prediction based on histopathological images using an adapted version of Inception v3 achieved an AUC between 0.733 and 0.856 for six frequently mutated genes [52]. Expanding beyond binary or paired classifications, Yang et al. (2021) developed a deep learning classifier based on EfficientNet-B5 capable of distinguishing six histopathological lung conditions, including lung adenocarcinoma, lung squamous cell carcinoma, small cell lung carcinoma, pulmonary tuberculosis, organizing pneumonia, and normal lung, from WSIs. The model demonstrated high concordance with ground truth labels and expert pathologists, with intraclass correlation coefficients exceeding 0.873 across all categories [56]. A recent meta-analysis reported pooled diagnostic accuracies of 93.5% (95% CI: 88.7–96.3) for subtyping renal cell carcinomas and 81% (95% CI: 67.8–89.6) for grading purposes [57]. In the context of immunohistochemistry, the deep learning model UV-Net demonstrated strong performance in evaluating Ki-67 proliferation index (PI) in breast cancer, significantly improving accuracy and inter-rater agreement among 90 international pathologists, while reducing the median turnaround time by 11.9% [58].

Building on research advances, several AI solutions have now transitioned from experimental to clinical deployment. For example, Galen™ Prostate Detect (© 2025 All Rights Reserved Ibex), an AI-powered solution for prostate cancer detection, received recently the FDA 510(k) clearance https://www.accessdata.fda.gov/scripts/cdrh/cfdocs/cfPMN/pmn.cfm?ID=K241232; accessed on 5 August 2025) [59]. In real-world clinical use, the system demonstrated a 99.6% positive predictive value (PPV) for cancer heatmap accuracy and uncovered 13% of cancers initially missed in a cohort of consecutive patients previously diagnosed as benign [59]. Clinical validation studies also indicate that AI can significantly improve workflow efficiency while maintaining non-inferior diagnostic accuracy compared to human pathologists. For instance, Paige Prostate Detect (https://www.paige.ai/diagnostic-ai; accessed on 5 August 2025) enhanced pathologist sensitivity by 7.3% and lowered diagnostic turnaround times by 65.5% in both simulated and real-world settings [60,61]. Similarly, validation of Paige’s AI for detecting breast lymph node metastases (https://info.paige.ai/breast; accessed on 5 August 2025) reported a 55% reduction in reading time and sensitivity improvements from 74.5% to 93.5% in two of three participating pathologists [62]. Moreover, the Paige PanCancer Detect tool (https://www.paige.ai/diagnostic-ai; accessed on 05 August 2025) is currently being piloted by NHS Wales to triage cases across multiple tissue types, further illustrating AI’s potential in routine diagnostic workflows [63]. Although adoption remains limited at present, these real-world implementations highlight how AI can already enhance efficiency and diagnostic safety in pathology workflows, bridging the gap between research innovations and clinical practice.

Task-specific deep learning models have also shown strong performance in non-oncologic settings. For instance, deep learning models were developed to differentiate between coeliac disease, nonspecific duodenitis, and normal tissue in H&E-stained slides [64]; to detect clinical heart failure from H&E-stained endomyocardial biopsies [65]; and to score total interstitial inflammation and peritubular capillaritis in kidney biopsies [66]. Although we do not detail machine learning performance metrics here, we emphasize their central role in model evaluation. We refer readers seeking a more technical understanding to the following excellent overviews [67,68].

In practice, the development and application of AI in pathology are deeply interwoven with human expertise. Supervised AI models are typically trained on large datasets annotated by expert pathologists, making pathologists essential not only in diagnosis but also in shaping the algorithm’s learning process. While highly effective in their respective domains, by enabling accurate and more homogenized evaluation of histopathological alterations, task-specific deep learning models are tailored to address narrowly defined problems. Their utility is limited by the need for extensive labeled data and retraining when applied to new diagnostic tasks or domains, which hinders scalability and generalizability.

General-purpose AI models

A paradigm shift in AI is occurring with the rise of general-purpose models termed “foundation models” that have been developed in order to overcome the scarcity of labeled data. Trained on vast and diverse datasets using generally self-supervision, these foundation models can be fine-tuned for a wide range of downstream tasks [69,70]. Current general-purpose models include GPT-4 (Generative Pretrained Transformer), BERT (Bidirectional Encoder Representations from Transformers), and CLIP (Contrastive Language-Image Pretraining) [71], among others. In pathology, foundation models are being developed and include vision encoders such as UNI [72], CTransPath [73], and Virchow [74], vision–language encoders such as PLIP [71], Prov-GigaPath [75], and CONCH [76], and, as we write this review, THREAD, a molecular-driven foundation model for oncologic pathology [77].

Most recently, a multimodal generative AI copilot for human pathology [78] named PathChat has been introduced. PathChat is an interactive tool with the ability to understand and respond to complex queries in natural language, hence facilitating intuitive interaction with end users. PathChat is capable of providing initial assessments by identifying potential histopathological features, suggesting a differential diagnosis, proposing ancillary testing and immunohistochemical (IHC) stains, and providing a final diagnosis, all in an interactive manner with a medical professional [78]. PathChat has been granted Breakthrough Device Designation by the U.S. Food and Drug Administration (FDA) (https://www.modella.ai/pathchat-fda-breakthrough-designation.html) (accessed on 5 August 2025). 

As technology matures over time, these foundational models will ultimately find impactful applications in clinical decision making. They also have significant potential for applications across medical education, preparing future pathologists to operate in an AI-driven ecosystem and research where AI facilitates the analysis of large-scale datasets, enabling discoveries in tumor biology, drug efficacy, and disease mechanisms. By combining advanced computational power with increasingly large datasets, AI models are capable of identifying patterns, highlighting anomalies, predicting molecular profiles, and even providing diagnostic insights with unmatched speed and accuracy. With routine tasks automation, such as tumor detection or quantification of biomarkers, AI not only accelerates workflows but also allows pathologists to focus on nuanced and complex cases that require clinical judgment and expertise. Moreover, AI-driven insights are opening new possibilities in personalized medicine, bringing the promise of tailoring therapies to individual patients with greater precision than ever before.

AI model limitations and challenges

Despite high-profile successes, numerous AI models in pathology continue to underperform when deployed under varied clinical conditions. Jarkman et al. (2022) demonstrated that a deep learning model for detecting breast cancer metastases in sentinel lymph nodes, trained on CAMELYON multicenter data, experienced a notable decline in performance when applied to a new diagnostic setting or to a small change in surgical indication, underscoring the challenges of model generalizability [79]. Vaidya et al. (2024) reported notable disparities in AI diagnostic accuracy across subgroups defined by race, insurance type, and age, showing that algorithms often performed significantly worse for underrepresented patients, an important equity concern [80]. More broadly, the systematic review by McGenity et al. (2023) covering 100 AI pathology studies revealed that many pathology AI tools suffer from a high risk of bias due to issues like lack of external validation, poor dataset diversity, and incomplete reporting of patient and imaging characteristics [81]. In dermatology, models evaluated with the Diverse Dermatology Images dataset showed 29–40% reductions in ROC-AUC when assessed on darker skin tones and uncommon diseases, demonstrating the real-world risks of demographic bias [82]. Likewise, performance degradation caused by domain shifts, such as variable staining, scanner models, and slide quality, has been repeatedly documented when used outside the development center [83]. These findings, among others, reinforce the necessity of rigorous external validation, use of representative and diverse training datasets, collaborative multi-institutional development, and the adoption of explainable AI frameworks as essential prerequisites for translating AI from promising prototypes into robust clinical tools [84].

## 3. The Role of the Pathologist: Today and Tomorrow

The evolving landscape of digital pathology and AI has raised fundamental questions about the role of pathologists, particularly in the diagnostic workflow. AI holds transformative potential, offering unprecedented precision, speed, and scalability. These advancements prompt us to assess how this technology impacts the clinical responsibilities of pathologists and whether it complements or challenges their central role in patient care. Ideally, the future of pathology should lie in a collaborative model, where human expertise and AI work in harmony to revolutionize diagnostics, research, and patient care. Pathologists must therefore adapt by embracing these tools while redefining their roles to remain central to diagnostic medicine. But this transition is not without uncertainty. As AI systems become more and more powerful, the profession must confront the possibility that tasks once considered core to pathology may be automated, thus raising important questions about identity, accountability, and the long-term relevance of the pathologist in the clinical workflow.

### 3.1. The Current Role of Pathologists and the Impact of AI on Pathology Workflow

Currently, pathologists play a multifaceted role in clinical medicine, including diagnostics, consultation, quality assurance, education, and research. Pathologists analyze tissue specimens, integrate findings with clinical data, and provide definitive diagnoses that guide treatment decisions. They also collaborate with multidisciplinary teams, offering expertise in complex cases and assisting in selecting appropriate ancillary tests. Moreover, pathologists oversee histopathology laboratory workflows to ensure compliance with regulatory standards and maintain diagnostic accuracy. In addition to diagnostic activity, pathologists contribute significantly to the training of medical students, residents, and fellows, to ensure the next generation is equipped with a robust understanding of histopathology. Finally, pathologists are inherently researchers, a role deeply rooted in their historical identity as scientist scholars [10]. Their ability to study diseases at the cellular and molecular levels makes them indispensable in advancing medical knowledge.

As AI is increasingly integrating into pathology workflows and continues to grow in sophistication and capability, its impact on these traditional roles must be critically examined. This can be summarized across two major dimensions: automation–augmentation and transformation. Automation–augmentation encompasses AI’s ability to perform and enhance diagnostic tasks such as detecting rare or subtle histopathological features, quantifying biomarkers, grading tumors, identifying patterns imperceptible to the human eye, and predicting disease outcomes and treatment response [51,85]. These capabilities enhance diagnostic accuracy and consistency, particularly in areas prone to interobserver variability [44,86,87]. While this improves performance and efficiency, over-reliance on AI in routine diagnostics may, however, lead to a loss of proficiency in certain techniques. Transformation, on the other hand, refers to the evolving role of pathologists as AI evolves. Instead of a frontline diagnostician, their role may shift toward that of a “diagnostic integrator” or “digital pathology consultant.” In this capacity, pathologists would synthesize AI-generated insights with clinical, molecular, and genomic data to provide comprehensive diagnostic reports. In the future, pathologists could act as strategic leaders in multidisciplinary teams, leveraging AI to personalize treatment plans and optimize patient care.

### 3.2. An Alternative Pathologist-Free AI-Assisted Diagnostic Workflow: Pathologist as Bystanders?

The integration of AI into pathology has largely been framed as an augmentative tool, one that enhances efficiency, accuracy, and workflow optimization while still requiring human oversight. However, as AI-powered models continue to evolve, a more disruptive question arises: Could AI eventually bypass pathologists altogether?

Recent advancements in foundation models (such as PathChat) and computer vision-based diagnostics have made it conceivable to envision a workflow in which AI, rather than a human pathologist, plays the central role in diagnosis. In this model, a fully automated pipeline could be established in which tissue samples are processed, digitized, analyzed, and interpreted without direct pathologist involvement. Instead, referring physicians, armed with sophisticated AI tools, would assume responsibility for diagnostic decision making (Figure 2).

To critically evaluate the feasibility of this paradigm, we first outline a hypothetical pathologist-free AI-assisted diagnostic workflow and then assess its advantages, limitations, and potential consequences.

In a pathologist-free AI-assisted diagnostic workflow, the diagnostic process could be restructured as follows (Figure 2):Specimen collection

The referring physician requests and coordinates the collection of a biopsy or a surgical specimen, which is sent to the anatomic pathology laboratory.

2.Grossing and tissue processing

Experienced laboratory staff perform gross examinations, tissue processing, slide preparation, and staining.

3.Slide scanning and digital conversion

Stained slides are scanned using slide scanners, generating high resolution whole-slide images (WSIs) that are stored on a local servers or on a cloud-based platform with associated metadata for downstream AI analysis.

4.AI-based analysis and interpretation

WSIs are transmitted to the referring physician, who engages with advanced AI diagnostic tools such as PathChat via natural language prompts. The AI system analyzes the WSIs, detects morphological patterns, quantifies biomarkers, and generates a preliminary diagnostic report. The physician reviews the findings and determines whether to proceed directly to treatment or request additional testing e.g., immunohistochemistry (IHC).

5.Final diagnosis and treatment plan

The physician integrates the AI-generated insights with results from any additional testing, as well as clinical, radiological, and molecular data to finalize the diagnosis. A treatment plan is then initiated based on this comprehensive, AI-assisted assessment.

While such a scenario may seem radical, it aligns with trends in radiology and dermatology, where AI-assisted diagnostics are already being explored for independent decision making. This raises fundamental concerns: Can AI sustain diagnostic accuracy without a pathologist’s oversight? What are the legal, ethical, and patient safety implications of minimizing pathologists’ roles in favor of automation?

A pathologist-free AI-assisted diagnostic workflow presents several advantages, including increased efficiency, faster turnaround time, reduced costs, and reduced interobserver variability. Moreover, by integrating histological, molecular, and clinical data, AI could support more precise, personalized medicine. However, despite these benefits, a pathologist-free AI-assisted diagnostic workflow has currently several limitations and challenges. AI models still lack the nuanced judgment of human pathologists, particularly in ambiguous or rare cases that can be misclassified. Moreover, most currently available AI models are not trained on sufficiently large, diverse, and, most importantly, across-population datasets, which raises concerns about AI bias and generalization issues across different patient populations [88]. Another major barrier with an AI-driven diagnosis is the so-called “black box” problem, which is associated with the lack of interpretability in deep learning models, making it difficult to understand how the AI model arrives at its decision [89,90]. Lack of understanding of the decision-making process poses relevant issues of liability. For instance, who will be responsible when an AI-based diagnosis leads to mismanagement or patient harm? Would a referring physician, without formal pathology training, be equipped to challenge or modify an AI’s decision? [91,92]. To build trust in an AI-standalone diagnosis, an understanding of its decision-making process is needed. Explainable AI and interpretable machine learning methods are currently a highly active field of research, with emerging solutions offering various degrees of interpretability of deep learning models [90]. The World Health Organization (WHO) issued AI ethics and governance guidelines for multimodal foundation models in 2024, outlining recommendations for governments and developers on responsible development and deployment (WHO, 2024).

The consequences of a shift toward a pathologist-free AI-assisted diagnostic workflow could be profound. The removal of human oversight could lead to a progressive deskilling of pathologists, as routine diagnostic experience is diminished, reducing expertise in handling complex cases. This transition could also alter workforce dynamics, requiring a redefinition of the pathologist’s role, shifting them from primary diagnosticians to AI supervisors and digital pathology consultants.

## 4. Future Directions

As AI models in pathology continue to evolve rapidly, the critical questions raised in this review on whether AI will augment, redefine, or replace the role of pathologists remain central. The scope of AI applications is not limited to simple task-specific models designed, for instance, for automated quantification of biomarkers, tumor staging, etc., but extends beyond to include highly sophisticated multimodal foundation models that can understand and respond to complex queries in natural language. These generative AI systems are capable of identifying histopathological features, suggesting differential diagnoses, proposing ancillary testing and immunohistochemical (IHC) stains, and providing a final diagnosis, all in an interactive manner with a medical professional [78]. In the near future, foundation models are expected to evolve into multimodal diagnostic assistants, capable of synthesizing histology, radiology, genomics, and clinical records into unified, structured reports. Early demonstrations, such as Prov-GigaPath and PathChat, already hint at this capability by assisting with complex differential diagnoses, recommending ancillary tests, and even drafting preliminary pathology reports autonomously. Such advancements are steering the field toward AI copilots that could one day manage most routine diagnostic tasks, shifting the role of human pathologists toward supervisory or consultative responsibilities.

Despite these advancements, there is currently limited attention to AI training in pathology education. How will training programs adapt to ensure future pathologists are equipped for an AI-driven landscape? Moreover, there is a lack of consensus on which new competencies pathologists may need to acquire, which skills need to be retained as AI takes over certain tasks, and what aspects of pathology can be entrusted to AI [93].

While AI is unlikely to completely replace pathologists in the “near future”, it will redefine their clinical role, emphasizing collaboration between human expertise and machine learning. The pathologist’s ability to adapt to these changes will determine their relevance in an AI-driven future [94]. As pathologists transition into an AI-driven era, three speculative outlooks, based on the extent of AI integration and the role of pathologists in decision making, emerge.

The first, a symbiotic model (most likely in the near term), in which AI complements pathologists by automating repetitive tasks such as pre-screening slides, highlighting regions of interest, or quantifying biomarkers, while they remain central to diagnosis and decision making.

The second is a transformational model (increasingly likely), where AI selectively offloads pathologists from certain routine, well-defined pathology tasks such as basic image interpretation, cell counting, tumor grading, and biomarkers quantification, shifting them from performing every step towards roles as digital pathology consultants and AI supervisors.

The third and most radical is a disruptive model. While unlikely in the immediate future, it becomes increasingly plausible in the long term as AI continues to evolve. In this scenario, AI reaches superhuman diagnostic accuracy, eliminating the need for pathologists’ oversight. In this case, referring physicians can leverage AI tools to generate diagnostic reports independently.

Looking ahead, as AI systems continue to improve in interpretability, generalizability, and real-time decision making, they may overcome many of the limitations outlined in the pathologist-free AI-assisted diagnostic workflow. If AI achieves full transparency, clinical reliability, and regulatory approval, it could fundamentally challenge the necessity of human pathologists in routine diagnostics. This trajectory underscores the provocative “pathologist-free” scenario proposed in this review, suggesting that the eventual replacement of human pathologists in certain domains may shift from speculative fiction to a plausible future reality.

The likelihood of this model materializing is significantly increased by the potential development of a domain-specific artificial general intelligence (AGI), which is an AI system capable of reasoning, adapting, and learning across a broad range of pathology tasks with minimal human input [95].

## 5. Conclusions

Galileo’s humble microscope has evolved into what many, at the dawn of the 21st century, considered “Science fiction”, now a cornerstone of modern pathology [96]. Digital pathology, combined with AI, has become a gateway to a new frontier, merging human expertise with machine learning to enhance diagnostic precision, drive groundbreaking research, and revolutionize medical education. By leveraging these advancements, the field has paved the way for a deeper understanding of disease mechanisms and more effective treatments.

As we stand at the intersection of technology and medicine, critical questions arise: What is the evolving role of pathologists in this rapidly advancing digital and AI-driven era? Will artificial intelligence merely augment their expertise or challenge their indispensability? And how can pathologists adapt to remain at the forefront of this transformative landscape? Moreover, could the development of more and more powerful AI tools mark a return of pathologists to their initial role of the mid-19th century as scientist scholars—focused on autopsy prosection, teaching, and theorizing about diseases—relegating them to roles outside direct clinical decision making?

Emerging multimodal AI systems, and potentially future domain-specific AGI, suggest that pathology may be approaching an inflection point that will fundamentally reshape their professional identity. Ultimately, the future of pathologists will be shaped not only by technological advancement alone, but also by how pathologists will choose to engage with it. Will they be passive bystanders or active architects of this inevitable transformation? These questions frame the ongoing dialogue about the role of pathologists today and in the future, as they navigate an evolving field redefined by innovation.

## Figures and Tables

**Figure 1 diagnostics-15-02308-f001:**
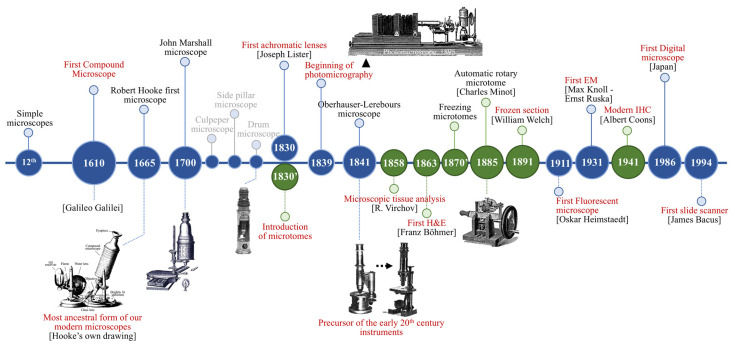
Timeline illustrating key historical milestones in microscopy and histotechnology that laid the foundation for digital pathology and AI implementation. Blue circles represent some of the major breakthroughs in microscopy from the early simple microscopes to the invention of the first compound microscope and later on the development of the first slide scanner. These include innovations such as achromatic lenses (Joseph Lister, 1830s), electron microscopy (EM; 1931), and the first slide scanner (1994, James Bacus). Green circles, highlight key discoveries in tissue processing and staining techniques which enhanced microscopic visualization and diagnosis, such as the introduction of microtomes (1830s), frozen sections (1891, William Welch), and modern immunohistochemistry (1941, Albert Coons). Embedded images of the photomicroscope and the automatic Minot microtome are adapted from Gal et al., 2001. Embedded images of Robert Hooke’s microscope and the Oberhauser-Lerebours microscope are from Kalderon, 1983. The image of John Marshall’s microscope is taken from the engraving titled “John Marshall’s New Invented Double Microscope for Viewing the Circulation of the Blood” published in Lexicon Technicum (circa 1704).

**Figure 2 diagnostics-15-02308-f002:**
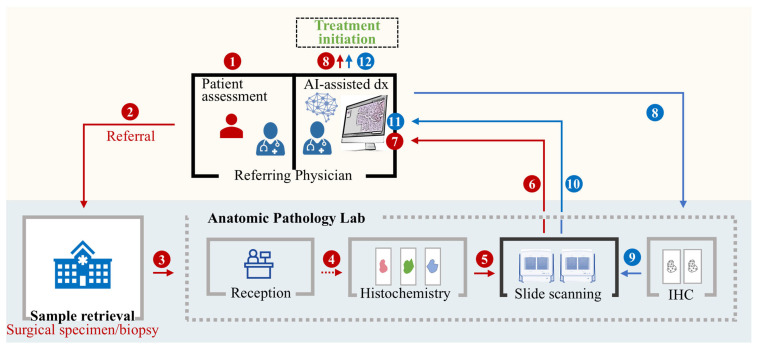
Pathologist-Free AI-Assisted Diagnostic Workflow. This figure illustrates a hypothetical AI-assisted diagnostic workflow, in which superior AI models replace the pathologist’s role, enabling the referring physician, e.g., an oncologist, to directly leverage AI tools for diagnosis. In this extreme scenario, AI-assisted tools generate diagnostic insights without a pathologist intervention, thereby challenging the traditional pathology workflow. Briefly, (1) the referring physician assesses the patient, (2) a referral is made for biopsy or surgical specimen collection, (3) the specimen is retrieved and sent to the anatomic pathology laboratory, (4) laboratory trained staff receive the case, perform gross examination, and process the tissue for histochemistry, (5) prepared and stained slides are scanned using, slide scanners to produce high-resolution digital images, (6) digital WSIs are transmitted back to the referring physician, (7) AI-powered diagnostic tools such as PathChat or other adapted platforms, analyze the WSIs, detect morphological patterns, quantify biomarkers, and generate a preliminary diagnostic report by interacting with the referring physician via natural language prompts. (8) The physician interprets the findings and determined whether to initiate a treatment or request additional testing e.g. immunohistochemistry (IHC). (9–10) If IHC is required, slides are prepared, scanned, and incorporated into the AI analysis, thereby providing additional results to the referring physician. (11) The AI system provides updated results to the referring physician. (12) Based on the complete AI-assisted diagnostic output, the referring physician finalizes the diagnosis and initiates treatment.

## Data Availability

No new data were created or analyzed in this study. Data sharing is not applicable to this article.

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
