# Peer review of "The Rise of AI-Assisted Diagnosis: Will Pathologists Be Partners or Bystanders?"

_diagnostics, 2025, doi:10.3390/diagnostics15182308_

Round 1

Reviewer 1 Report

Comments and Suggestions for Authors

A remarkable review study that questions where pathologists will stand in this situation as artificial intelligence becomes widespread and takes its place in diagnostic systems. From this perspective, it is thought to contribute to the literature. 

However, some corrections are needed in order to attract the reader's attention. These are;

1- In Section 2.2 Artificial Intelligence: A New Frontier in Pathology. Task-specific deep learning models should be detailed. This section has been passed very superficially.

2- It remains unclear which criteria were taken into account when creating the literature. In order to be more explanatory, it is not clear which period and which criteria the literature was determined according to. These should be explained.

3- What was the force that motivated the authors to do this study? Was there a gap in the literature in this direction? The motivation should be fully detailed.

4- It was seen that the expression "is too late?" was used in the abstract section of the article. This use is not appropriate. It is recommended that this sentence be changed.

5- If more studies on unsuccessful or low-performing artificial intelligence-based diagnostic systems are added, this article will mature more. Research should be done on this subject and unsuccessful examples should be added.

6- It has been observed that the methods of diagnosing diseases of artificial intelligence-based systems are generally categorized. However, examining the metrics by which the performance of these AI methods is evaluated (accuracy, f1_score, precision, recall, etc.) will contribute to the maturation of the article.

Author Response

Comment 1: In Section 2.2 Artificial Intelligence: A New Frontier in Pathology. Task-specific deep learning models should be detailed. This section has been passed very superficially.

Response1: We thank the reviewer for this insightful suggestion. We agree that this section can be enhanced by including few examples of specific task-specific deep learning models along with their applications and performance metrics. Accordingly, section 2.2 was amended to briefly describe representative task-specific AI models with their performance measures.

Comment 2: It remains unclear which criteria were taken into account when creating the literature. In order to be more explanatory, it is not clear which period and which criteria the literature was determined according to. These should be explained.

Response 2: We appreciate the reviewer’s remark. We conducted a comprehensive literature review using mainly PubMed, Web of Science, and Google Scholar databases, employing Key search terms such as ‘digital pathology’, ‘artificial intelligence’, ‘deep learning’, ‘computational pathology’, ‘pathologist role’, and ‘AI-driven diagnostics’. In our search we prioritized original articles, systematic reviews, and authoritative opinion pieces published primarily between 2010 and late 2024/early 2025. Selected articles were based on relevance to digital pathology technology, AI methodologies, clinical impact, educational implications, and ethical and regulatory considerations. Sources were restricted primarily to peer-reviewed literature.

Since our manuscript is a conceptual review rather than a systematic one, and therefore without a methods section, we did not include these selection criteria explicitly in the main text. However, we hope this clarification adequately addresses reviewer concern regarding our approach to literature selection.

Comment 3: What was the force that motivated the authors to do this study? Was there a gap in the literature in this direction? The motivation should be fully detailed.

Response 3: The abstract was amended to explicitly state and clarify our motivation behind this manuscript.

Comment 4: It was seen that the expression "is too late?" was used in the abstract section of the article. This use is not appropriate. It is recommended that this sentence be changed.

Response 4: This expression in the abstract is now revised.

Comment 5: If more studies on unsuccessful or low-performing artificial intelligence-based diagnostic systems are added, this article will mature more. Research should be done on this subject and unsuccessful examples should be added.

Response 5: We appreciate this valuable suggestion. Indeed, despite impressive achievements, several AI-based systems have encountered significant practical challenges. In response, we have expanded section 2.2 by introducing a new subsection entitled “ AI-models limitations and challenges” where we explicitly incorporate concrete examples from the literature to highlight common pitfalls, such as poor generalization, suboptimal accuracy, and interpretability issues.

Comment 6: It has been observed that the methods of diagnosing diseases of artificial intelligence-based systems are generally categorized. However, examining the metrics by which the performance of these AI methods is evaluated (accuracy, f1_score, precision, recall, etc.) will contribute to the maturation of the article.

Response 6: We thank the reviewer for this thoughtful observation. We fully agree that evaluation metrics such as accuracy, precision, recall, F1-score, and AUC play a crucial role in assessing the performance of AI models and are foundational to understanding their strengths and limitations.

However, after careful consideration, we believe that a more detailed examination of how each metric functions may be more appropriate for a methods-focused or technical review, rather than the present manuscript, which aims to provide a broader conceptual and critical overview of digital pathology and AI applications. Our goal is to reach a diverse audience, including clinicians, trainees, and researchers, and to keep the article accessible and focused on clinical and translational relevance.

That said, we have revised section 2.2. to briefly acknowledge the importance of these metrics and have directed interested readers to dedicated and excellent reviews that explain clearly  their technical foundations and clinical interpretation in the context of medical AI.

Reviewer 2 Report

Comments and Suggestions for Authors

Dear Authors! Thank you for pssibility to review your manuscript

Now you manuscript is lokks like a book chapter, please re-write it according the standards of the review manuscript and decrease the well-know historical aspects and focus on the goals of the manuscript

Author Response

Comment1: Now you manuscript is lokks like a book chapter, please re-write it according the standards of the review manuscript and decrease the well-know historical aspects and focus on the goals of the manuscript

Response 1: We thank the reviewer for suggesting that we align our manuscript more closely with the standards of a review article. Although many readers may indeed be familiar with the general history of microscopy, we believe this background remains essential for contextualizing the rapid changes and innovations in digital pathology today. Moreover, we've found that many pathologists, pathology residents, researchers, and interdisciplinary scientists often lack knowledge of critical historical developments that underpin current technological advances. Presenting this background in a brief, structured, engaging, and narrative-driven manner allows us to connect the past to the present and the future in a more impactful way.

Most importantly, this concise historical overview directly informs our central question regarding the potential future role of pathologists. Specifically, by reviewing how pathologists transitioned historically from scientist-scholars in the mid-19th century to their current specialized diagnostic roles, we contextualize and critically assess our proposed alternative AI-driven, "pathologist-free" workflow. This historical framing thus naturally leads to our provocative exploration of whether emerging AI technologies might cause the discipline to revert, in some respects, back toward the earlier scholarly and research-oriented role that defined pathology practice more than a century ago.

In response to the reviewer’s suggestion, we have carefully revised and moderately shortened the historical sections (particularly Section 1.3) to improve conciseness and readability. Additionally, we have enhanced the focus on the manuscript’s central thesis throughout, thereby aligning it more clearly with the expectations of a scholarly review article.

Reviewer 3 Report

Comments and Suggestions for Authors

The manuscript analyses the relevance of artificial intelligence applied to immunohistochemistry as a diagnostic tool in association/substitution of a pathologist.

This topic is of great interest nowadays. 

The authors made a brief history of the steps leading to the whole slide analysis with a scanner and the possible application of AI. Even, they consider the possibility of using this tool instead of a pathologist.

The description is well organized, and the reader can understand well the point of view of the authors. 

This review does not describe/report the experimental data from papers in which the pathologist analyses and the digital pathology (with or without the help of the AI) are compared. This topic aims to characterize the level of accuracy, precision, and reproducibility of data obtained on a given cohort of patients in retrospective or prospective studies.

This reviewer used the whole scan of slides of immunohistochemistry mainly for research purposes. This analysis can be easily applied to diagnostic procedures. Also, the training sets of slides are analyzed by pathologists to teach the scanner.  The AI is based on the pathologist's analysis at least at the beginning. It would be nice to have discussed the differences between AI and pathologists in a practical setting, putting in evidence actual instances in which these two approaches have been used.

A brief analysis of the instruments for digital pathology present on the market (Leica, 3DHistech, Hamamatsu, and so on), with a detailed description of the pros and cons among them, could help the reader to better understand the relevance of digital pathology to a wider audience than simple pathologists. Also, the software used by the different systems on the market and the use of some of them as actual diagnostic tools should be considered.

The authors focused mainly on AI, but I think that digital pathology is the intermediate step on which AI is applied.

Author Response

Comment1: This review does not describe/report the experimental data from papers in which the pathologist analyses and the digital pathology (with or without the help of the AI) are compared. This topic aims to characterize the level of accuracy, precision, and reproducibility of data obtained on a given cohort of patients in retrospective or prospective studies.

Response 1: We thank the reviewer for highlighting this. While our review is conceptual rather than systematic, we agree adding a short overview comparing AI and human pathologist performance would significantly enhance our manuscript. A short subsection summarizing representative comparative studies of pathologist versus AI performance clearly noting AI performance compared to human experts is now added to section 2.2 subsection “task-specific AI models”.

Comment2: This reviewer used the whole scan of slides of immunohistochemistry mainly for research purposes. This analysis can be easily applied to diagnostic procedures. Also, the training sets of slides are analyzed by pathologists to teach the scanner.  The AI is based on the pathologist's analysis at least at the beginning. It would be nice to have discussed the differences between AI and pathologists in a practical setting, putting in evidence actual instances in which these two approaches have been used.

Response 2: We appreciate the reviewer’s thoughtful observation regarding the practical setting differences between AI and pathologists, particularly regarding the crucial role pathologists currently play in training and validating AI models. We fully agree this important point which is now highlighted in section 2.2 subsection “task-specific AI models”. we explicitly stressed how AI relies heavily on pathologist-derived "ground truth" annotations and how these initial human inputs significantly influence AI accuracy and utility. We also added a practical comparative study that illustrates the interplay between AI results and pathologist interpretation.

Comment 3: A brief analysis of the instruments for digital pathology present on the market (Leica, 3DHistech, Hamamatsu, and so on), with a detailed description of the pros and cons among them, could help the reader to better understand the relevance of digital pathology to a wider audience than simple pathologists. Also, the software used by the different systems on the market and the use of some of them as actual diagnostic tools should be considered.

Response 3: We sincerely thank the reviewer for this valuable suggestion. We fully agree that a comparative analysis of commercially available digital pathology scanners and software systems, including platforms by Leica, 3DHistech, Hamamatsu, and others, would be highly informative for readers from diverse backgrounds. However, given the conceptual and integrative focus of the current review, which aims to examine the evolving role of artificial intelligence in pathology rather than provide a technical assessment of specific hardware or software solutions, we believe that a detailed device-by-device comparison may be more appropriate for a methods-focused or technology-driven review.

That said, we recognize the importance of this topic and have prompted, in subsection 1.5, interested readers to several excellent and up-to-date reviews that provide comprehensive analyses of slide scanner technologies, diagnostic validation, software tools, and platform comparisons. We believe these resources will be particularly useful to readers seeking a deeper understanding of available systems and their clinical implementation potential.

Comment 4: The authors focused mainly on AI, but I think that digital pathology is the intermediate step on which AI is applied.

Response 4: We completely agree, and we now clearly articulated this distinction in the introductory part of section 2, emphasizing digital pathology as foundational technology enabling AI integration.

Round 2

Reviewer 2 Report

Comments and Suggestions for Authors

Dear Authors!

Thank you for the corrections done in the manuscript.

The manuscript might be intersting for readerds but it does not meet the criteria of research/scientific manuscript. It is a real description.